# Impact of COVID-19 pandemic on surgical neuro-oncology multi-disciplinary team decision making: a national survey (COVID-CNSMDT Study)

Stephen John Price [1] ,[1] Alexis Joannides,[1] Puneet Plaha,[2]
Fardad Taghizadeh Afshari,[3] Erminia Albanese,[4] Neil U Barua,[5] Huan Wee Chan,[6]
Giles Critchley,[7] Thomas Flannery,[8] Daniel M Fountain,[9] Ryan K Mathew [1] ,[10]
Rory J Piper,[2] Michael TC Poon,[11] Chittoor Rajaraman,[12] Ola Rominiyi,[13]
Stuart Smith,[14] Georgios Solomou,[3,15] Anna Solth,[16] Surash Surash,[17]
Victoria Wykes,[18] Colin Watts,[19] Helen Bulbeck,[20] Peter Hutchinson,[1]
Michael D Jenkinson,[21] On behalf of the COVID-CNSMDT study group

For numbered affiliations see end of article.

**Correspondence to**
Stephen John Price;
sjp58@cam.ac.uk

## ABSTRACT

**Objectives** Pressures on healthcare systems due to COVID-19 has impacted patients without COVID-19 with surgery disproportionally affected. This study aims to understand the impact on the initial management of patients with brain tumours by measuring changes to normal multidisciplinary team (MDT) decision making.

**Design** A prospective survey performed in UK neurosurgical units performed from 23 March 2020 until 24 April 2020.

**Setting** Regional neurosurgical units outside London (as the pandemic was more advanced at time of study).

**Participants** Representatives from all units were invited to collect data on new patients discussed at their MDT meetings during the study period. Each unit decided if management decision for each patient had changed due to COVID-19.

**Primary and secondary outcome measures** Primary outcome measures included number of patients where the decision to undergo surgery changed compared with standard management usually offered by that MDT. Secondary outcome measures included changes in surgical extent, numbers referred to MDT, number of patients denied surgery not receiving any treatment and reasons for any variation across the UK.

**Results** 18 units (75%) provided information from 80 MDT meetings that discussed 1221 patients. 10.7% of patients had their management changed—the majority (68%) did not undergo surgery and more than half of this group not undergoing surgery had no active treatment. There was marked variation across the UK (0%–28% change in management). Units that did not change management could maintain capacity with dedicated oncology lists. Low volume units were less affected.

**Conclusion** COVID-19 has had an impact on patients requiring surgery for malignant brain tumours, with patients receiving different treatments—most commonly not receiving surgery or any treatment at all. The variations

## Strengths and limitations of this study

► This is a national survey that covers 75% of the UK units outside London providing a view of the impact COVID-19 had on the management of patients with brain tumours across the UK.
► Our data are based on 1221 patients.
► Data collection occurs 2 weeks before and 2 weeks after the peak of COVID-19 infections.
► We have not collected patient-level data so can't assess what happens to individual patients.
► We have only looked at surgical management and do not have data on oncology treatment.

show dedicated cancer operating lists may mitigate these pressures.

**Study registration** This study was registered with the Royal College of Surgeons of England's COVID-19 Research Group (https://www.rcseng.ac.uk/coronavirus/ rcs-covid-research-group/).

## BACKGROUND

The COVID-19 pandemic has caused an unprecedented threat to healthcare delivery worldwide. Hospitals dealing with large numbers of patients requiring critical care have redeployed staff and converted operating rooms into intensive care units to cope with infected patients. This has had a marked impact on services and patient care for patients without COVID-19. Surgery has been disproportionately affected and requests to maintain urgent cancer surgery requires careful patient triage.[1 2]

In the UK adult brain tumours are managed across 30 regional neurosurgical centres, 24 of these are outside London. All new brain diagnoses are discussed in a multidisciplinary team (MDT) meeting that decides the optimal treatment for each patient. Each MDT will review a range of tumours from non-malignant or slow-growing (low-grade) tumours whose initial treatment could be safely deferred, to the malignant tumours that are characterised by rapid growth and patient deterioration without treatment in a matter of weeks. Surgery is the main initial treatment for most patients—it can cure non-malignant tumours, remove the bulk of malignant tumours and is the only way of providing tumour tissue for pathological and molecular diagnosis to guide further treatment. Although surgery is not curative in malignant tumours, it does prolong good quality survival. National guidelines recommended that surgery for malignant tumours should continue during the COVID-19 pandemic providing adjuvant oncology treatment is available.[3] The role of the MDT is critical to good patient management.

We undertook a national survey to evaluate the impact of COVID-19 on neurosurgical oncology services and to explore differences and variations in MDT decisions compared with the pre-COVID-19 era to investigate if patients were denied access to surgery.

## METHODS

Project leads from each of the UK neurosurgical units were identified to prospectively collect information from their weekly neuro-oncology MDT between 23 March and 20 April 2020. London units were excluded as the pandemic peak was more advanced at the time of this study.

The primary outcome measure in this study is the change in MDT decision making. This was determined by each MDT. At each MDT meeting, the project lead would review the decisions made for every patient with the rest of the team and decide whether this decision had differed from their normal practice due to COVID-19.

At each MDT meeting, the following data were recorded:

► Total number of new patients discussed.
► Number of patients where the MDT felt their initial/surgical treatment was different from their standard management due to COVID-19.
► Number of patients who did not receive surgery or whose surgical treatment intent was changed (eg, biopsy rather than resection).
  – From the group that did not undergo surgery, the number of the patients who did not receive any active treatment (ie, supportive care only).
► Number of patients who had undergone surgery for a malignant tumour (defined as high-grade glioma, metastatic tumour or other, rarer malignant tumour).

## Variations in decision making

Data were compared with a baseline MDT workload from the 4 weeks of February (3–28 February 2020) to provide information on the impact of COVID-19. To explore the variations in MDT decision making, funnel plots for the proportion of cases discussed where management was altered were generated using Microsoft Excel, with 95% and 99.8% control limits calculated using the Wilson Score method for binomially distributed variates.[4] Project leads were invited to comment with free text on how their unit responded to COVID-19. Using this free text, we developed three groups that arranged units based on commonly expressed themes of how they managed to mitigate the challenges of COVID-19.

## Statistical analysis

Statistical analysis was performed using IBM SPSS V.26. All variables were assessed for normality using a 1-sample Kolmogorov-Smirnov test. Changes in the weekly mean number of new patients discussed at MDT meetings and operations for malignant tumours was assessed using paired t-test. Correlation of these variables with changes of MDT decision rates was made using a Pearson correlation.

## Patient and public involvement

Patients/the public were not involved in the concept or design of this study as this came about from concerns among professionals at the services they were providing patients. A patient/public representative was appointed to the study group during the project set up to comment on the results.

## RESULTS

Eighteen of the 24 eligible neurosurgical units (75%) participated in the study. A total of 1221 new patients were included from 80 MDT meetings. In all units, the functioning of the MDT changed with eight units (33%) moving to fully video conferencing, all of the other units would limit attendance to a smaller number of senior representatives of different specialities (eg, neurosurgery, oncology, radiology, etc). Four units (17%) reduced the duration of the meeting.

Comparing activity to baseline, there was a significant reduction in the number of new patients discussed from a mean weekly baseline table 1. In total, 131 patients (10.7%) had a change in their initial management due to COVID-19. Of these, 15 (11.5%) had a change of surgical intent, and 90 (68%) had no surgery at all. Forty-seven patients (52% of those decided for no surgery) were decided for best supportive care. Information from project leads suggest the majority of these patients were elderly and had poor performance status. Other reasons for not operating were patients with low-grade gliomas or meningiomas whose surgery was deferred until after the critical phase of the pandemic.

**Table 1** Changes in workload between baseline and COVID-19 period. Malignant tumours refer to high-grade gliomas, metastases and other rarer malignant tumours

| | Baseline | During study period | Significance | Difference |
|---|---|---|---|---|
| Number of new patients | 21.2 (95% CI 14.0 to 28.3) | 15.3 (95% CI 10.0 to 20.6) | t=−4.26, p=0.0005 | ↓27% |
| Number of operations for malignant tumours | 4.1 (95% CI 3.2 to 5.0) | 3.0 (95% CI 2.0 to 4.1) | t=−2.76, p=0.01 | ↓26% |

The change in initial management varied between centres and ranged from 0% to 28%. This did not correlate with either the change in the number of weekly operations (R=−0.03; p=0.91) nor the number of patients discussed at an MDT meeting (R=−0.29; p=0.29). The funnel plot identified three groups of units (figure 1). Group A was smaller units with low numbers of new patients with brain tumour (mean=7.4, 95% CI 4.1 to 10.6) and low rates of management changes. Project leads commented that COVID-19 had less impact on their units or were able to cope with the numbers of referrals. The group included a unit whose numbers decreased from 28.5 to 13.6 per week (52% reduction). Group B was larger units (mean weekly new patients=34.1, 95% CI 20.9 to 47.2) that allocated dedicated oncology lists. This was achieved by a variety of methods—implementing daily oncology lists, managing patients as emergencies or using the private hospital sector to provide space to treat brain tumour patients. Group C was the group with the largest change in management. A common theme for this group was the lack of dedicated oncology lists—patients with malignant tumours competing for surgical theatre operating time with other urgent surgical cases.

### Patient/public interpretation of results

The results were examined by the study group's patient/public representative (HB). They commented that "when you're diagnosed with a brain tumour, your whole life is not only upended but put on hold. At this point, patients and their families need certainty. To be denied surgery—the first-line treatment—doesn't bear thinking about. This impacts on the whole of the patient pathway, as without biopsies and resections, nobody really knows what you are treating, or how much tumour you are treating. It undermines the whole of the personalised medicine agenda and this cohort of patients will be living with the consequences, but probably not for as long as they would have been if they had had surgery. The short-sighted decision to bypass brain tumour surgery is undoubtedly shortening patients' lives."

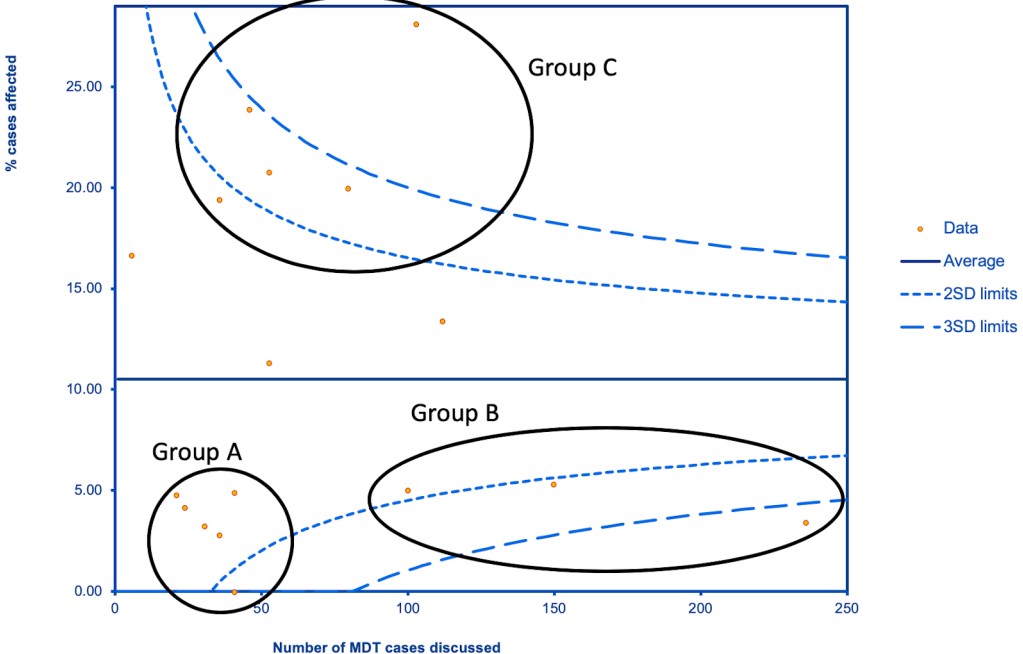

**Figure 1** Funnel plot showing variation in number of patients whose management was changed due to COVID-19 versus the number of patients discussed in MDTs over the study period. SD curves have been included. we have broken the data into three groups with different responses to the pandemic.

## DISCUSSION

This national study has shown the impact COVID-19 has had on MDT decision making for malignant (gliomas and metastases) brain tumour patients. In total, 10.7% of patients had their initial management changed, of which the majority (68%) did not undergo surgery. Over half of these patients had palliative care only. Surgery is the first-line treatment in malignant brain tumours and extent of resection is the only modifiable prognostic factor.[5] Patients with glioblastomas, the most common malignant primary tumour, have a very limited survival without treatment,[6] and surgical resection improves the efficacy of adjuvant therapies and quality of life.[7]

Like other cancers, COVID-19 has impacted on referrals.[8] We found a 27% reduction in a number of patients discussed and operations performed for malignant tumours (26% reduction) (cf. table 1). Initial symptoms of brain tumours are frequently non-specific, and patients often present when they decompensate.[9 10] The reduction in new tumour referrals during this study may result in patients presenting later and with more extensive disease.

An interesting finding was the variation between units. Group analysis identified trends in units that successfully reduced the impact of COVID-19 on MDT management. Strategies to ensure that dedicated cancer lists continued were helpful in larger units. It is clear that where cancer patients were scheduled on operating lists with other urgent cases, more substantial MDT management changes were needed to cope. Our findings complement other studies that have shown that the use of the private sector during this COVID-19 pandemic results in less disruption to services.[11]

The study does have limitations, principally in the design. The purpose was to provide a prospective, rapid 'snapshot' of changes of MDT decision making during the peak of the pandemic using high-level data. As such, we did not include the detail from individual patient data that would perhaps allow a better explanation of individual variation between units. Instead, our project leads, all busy clinicians dealing with the impact of the peak of the pandemic, were able to obtain data on patient numbers. Although the study may lack some scientific rigour, it nevertheless provides a narrative of the real-world experiences seen by neurosurgical units during the height of the pandemic in the UK. Although this survey covered 70% of units, not all geographical areas have been included (eg, South West and London). We have not been able to include the COVID-19 infection rate across the country into our analysis since each neurosurgery unit serves a large geographical area. The dates of our survey, however, correspond to the peak in cases for every geographical region.

Our primary outcome measure was, by its nature, pragmatic. This introduces some subjectivity between units as to what they defined as a change in management. As most of these changes involved no surgery or a change in surgical intent, we believe project leads reported significant changes in management that this study sought to explore. Our study only analysed changes in surgical treatment, but we know there are also changes in adjuvant therapy, such as radiation dose and chemotherapy regimes. It was difficult to quantify these changes in our study since patients are often treated in oncology centres that are separate from the neurosurgery centres. The main focus of our study was to look at the MDT decision making where decisions about adjuvant therapy were made to influence surgery as national guidelines recommend surgery for patients with malignant tumours only where adjuvant therapy was available.

In conclusion, COVID-19 has had an impact on patients requiring surgery for malignant brain tumours—with 10.7% having a change of management due to COVID-19, most commonly by not having surgery and many having no active treatment at all. The variation of changes in decision making shows that dedicated cancer operating lists may help to mitigate the pressures of COVID-19.

**Author affiliations**
[1]Neurosurgery Division, Department of Clinical Neurosciences, Cambridge University, Cambridge, UK
[2]Department of Neurosurgery, John Radcliffe Hospital, Oxford, UK
[3]Department of Neurosurgery, University Hospitals Coventry and Warwickshire National Health Service Trust, Coventry, UK
[4]Department of Neurosurgery, University Hospitals of North Midlands National Health Service Trust, Stoke-on-Trent, UK
[5]Department of Neurosurgery, North Bristol National Health Service Trust, Bristol, UK
[6]Department of Neurosurgery, University Hospital Southampton National Health Service Foundation Trust, Southampton, UK
[7]Department of Neurosurgery, Brighton and Sussex University Hospitals National Health Service Trust, Brighton, UK
[8]Department of Neurosurgery, Royal Victoria Hospital, Belfast, UK
[9]Manchester Centre for Clinical Neurosciences, Salford Royal National Health Service Foundation Trust, Salford, UK
[10]Department of Neurosurgery, Leeds General Infirmary, Leeds, UK
[11]Usher Institute, The University of Edinburgh, Edinburgh, UK
[12]Department of Neurosurgery, Hull Royal Infirmary, Hull, UK
[13]Department of Neurosurgery, Royal Hallamshire Hospital, Sheffield, UK
[14]Department of Neurosurgery, Queen's Medical Centre, Nottingham, UK
[15]College of Medical and Dental Sciences, University of Birmingham, Birmingham, UK
[16]Department of Neurosurgery, Ninewells Hospital, Dundee, UK
[17]Department of Neurosurgery, Royal Victoria Infirmary, Newcastle upon Tyne, UK
[18]Department of Neurosurgery, Queen Elizabeth Hospital, Birmingham, UK
[19]Institute of Cancer and Genomic Studies, University of Birmingham, Birmingham, UK
[20]Brainstrust, Cowes, UK
[21]Department of Neurosurgery, The Walton Centre National Health Service Foundation Trust, Liverpool, UK

**Collaborators** COVID-CNSMDT Study Group: Nicola Johnson, Yasir Chowdhury, Mandy Lynch, George Malcolm, Ven Iyer, Constantinos Charalambides, Chris Herbert, Richard Mair, Thomas Santarius, Fiona P Harris, Olivia Poulter, Sandeep Solanki, Kismet Hossain-Ibrahim, Paul Brennan, Shailendra Achawal, Gerry O'Reilly, John Goodden, Simon Thomson, Andrew R Brodbelt, Emmanuel Chavredakis, David DA Lawson, Jibril Farah, Tina Karabatsou Syed Shumon, Damian Holliman, Muhammed Zafar, William Sage, Murugan Sitaraman, Yahia Al-Tamimi, Paul Grundy, Ebere Ogbonnaya.

**Contributors** SJP, MDJ, PP, CW and PH were involved in the study concept and design. SJP, TF, VW, GC, NUB, FTA, AS, MTP, CR, RKM, MDJ, DMF, SS, SSu, GS, RJP, OR, HWC and EA were involved in collecting data from each site and guarantee the data at each site. SJP and AJ were involved in data analysis. HB

provided the patient perspective on our findings. SJP wrote the first draft of the paper and all authors were involved in revising the paper and have approved the final paper.

**Funding** SJP is funded by a National Institute for Health Research (NIHR), (Career Development Fellowship).This paper presents independent research funded by the National Institute for Health Research (NIHR). The views expressed are those of the author(s) and not necessarily those of the NHS, the NIHR or the Department of Health and Social Care. AJJ is funded by the NIHR Brain Injury MedTech Co-operative.MTCP is supported by Cancer Research UK Brain Tumour Centre of Excellence Award (C157/A27589).

**Competing interests** None declared.

**Patient consent for publication** Not required.

**Ethics approval** Ethical approval was not required as no patient-specific information was collected.

**Provenance and peer review** Not commissioned; externally peer reviewed.

**Data availability statement** Data are available upon reasonable request. Data access queries should be addressed to the corresponding author (SJP). All data is anonymised and individual unit level data is not available. Approval for data release will be made by SJP, MDJ, PP and PH who conceived the study.

**ORCID iDs**
Stephen John Price http://orcid.org/0000-0002-7535-3009
Ryan K Mathew http://orcid.org/0000-0002-2609-9876

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
