## [Reviewer comments · BMJ Open]

ARTICLE DETAILS

TITLE (PROVISIONAL)	Impact of COVID-19 Pandemic on Surgical Neuro-oncology Multi-Disciplinary Team Decision Making – A National Survey (COVID-CNSMDT Study)
AUTHORS	Price, Stephen; Joannides, Alexis; Plaha, Puneet; Afshari, Fardad; Albanese, Erminia; Barua, Neil; Chan, Huan Wee; Critchley, Giles; Flannery, Thomas; Fountain, Daniel; Mathew, Ryan; Piper, Rory; Poon, Michael; Rajaraman, Chittoor; Rominiyi, Ola; Smith, Stuart; Solomou, Georgios; Solth, Anna; Surash, Surash; Wykes, Victoria; Watts, Colin; Bulbeck, Helen; Hutchinson, Peter; Jenkinson, Michael

VERSION 1 – REVIEW

REVIEWER	Luiz P Kowalski University of Sao Paulo Medical School and A C Camargo Cancer Center, Brazil
REVIEW RETURNED	10-Jun-2020

GENERAL COMMENTS	This is a well written paper. The objective is clear and the method used is appropriate to get the information needed during a pandemic period. The results are analysed properly and the discussion, including study limitations well presented. Conclusions are compatible with the method and results achieved.
--

REVIEWER	René H.M. Verheijen UMC Cancer Centre, Utrecht Dept. of Gynaecological Oncology The Netherlands
REVIEW RETURNED	16-Jun-2020

GENERAL COMMENTS	This paper attempts to quantify changes in decision making regarding the surgical treatment (only) of patients with malignant brain tumours during the (peak) time of the COVID-19 pandemic. The main concern regards the primary outcome, as this is defined as the management decision change as defined by the MDT (p.9 l.27). To what extent and how MDT came to the conclusion that management was changed as compared to standard treatment is not clarified, nor quantified. A quantitative comparison has been made with the 'baseline workload', but not with the baseline decisions/indications for surgery etc. In order to support the conclusions and the interpretation (especially by the patient support group) the authors could perhaps elucidate or even quantify how and to what extent this change in decision was determined.
--

	Minor points:  - p.11, l.27 there is no correlation found between changes in initial management and neither number of patients discussed, nor with number of operations. A bird's eye view of table 1, however, does at least suggest a correlation between the latter two, thus suggesting that the decrease in number of operations is a (logical) result of a decrease of patients presented. Was the 10% change in decision too small to be reflected in a change in operations ? Other explanations ? - it is suggested that, at least for Group B, use of the private sector could achieve allocation of sufficient oncology lists. It would be interesting to see to what extent this solution was used and could the authors explain whether only non-oncological patients were referred to the private sector or also oncological patients ? This should also be discussed in the wake of data from the COVID-19 period indicating that cancellations were less likely to occur in the private sector (Jean e.a., Acta Neurochir 7th April 2020). - Finally, could the authors speculate or quantify on the reasons why no surgery was performed in the 90 cases in whom this normally would have been recommended ? Did additional factors like age or condition also play a factor, was this decision always made because of lack of theatre capacity?
--	---

REVIEWER	Kiran Turaga University of Chicago USA
REVIEW RETURNED	17-Jun-2020

GENERAL COMMENTS	The authors provide an analysis of neurosurgical units and describe the change in the MDT management for cancer patients due to the COVID crisis. They further break down units based on the number of cases or competing interests and find that the differences in management were further amplified. This is an interesting paper, but the research could have delved further into the question at hand.  1. For instance, if the management was changed in 10% of patients, did that lead to different outcomes for patients, or were they identical? Given the short follow up, I understand not having survival outcomes, but perhaps resectability or completeness of resection rates? 2. I think clear understanding of local factors is important to understand the implications of the analysis. Firstly, were MDTs conducted virtually or in person at all institutions? Were all physicians involved or were certain demographics of physicians excluded (eg >65)? Was there an association between testing rates, covid positivity rates, healthcare worker positivity, ICU bed shortage and PPE usage to the recommendation? The authors comment for instance group A had less covid cases, but data would be welcome. 3. Are all patients managed according to a pre-specified pathway and there were 10% deviance from the pathway, or was this a subjective assessment of change from what might have been done if not for COVID? 4. Can the group include the neurosurgical units from London too? This would give them a control of practices in a higher incidence zone and may support their suggestion that a lower incidence of covid leads to less change.
---

VERSION 1 – AUTHOR RESPONSE

Reviewers Comments to Author:

Reviewer: 1

Reviewer Name: Luiz P Kowalski

Institution and Country: University of Sao Paulo Medical School and A C Camargo Cancer Center, Brazil

Please state any competing interests or state 'None declared': None.

This is a well written paper. The objective is clear and the method used is appropriate to get the information needed during a pandemic period. The results are analysed properly and the discussion, including study limitations well presented. Conclusions are compatible with the method and results achieved.

Thank you for your kind comments.

Reviewer: 2

Reviewer Name: René H.M. Verheijen

Institution and Country: UMC Cancer Centre, Utrecht Dept. of Gynaecological Oncology, The Netherlands

Please state any competing interests or state 'None declared': None declared

This paper attempts to quantify changes in decision making regarding the surgical treatment (only) of patients with malignant brain tumours during the (peak) time of the COVID-19 pandemic.

The main concern regards the primary outcome, as this is defined as the management decision change as defined by the MDT (p.9 l.27). To what extent and how MDT came to the conclusion that management was changed as compared to standard treatment is not clarified, nor quantified. A quantitative comparison has been made with the baseline workload, but not with the baseline decisions/indications for surgery etc.

In order to support the conclusions and the interpretation (especially by the patient support group) the authors could perhaps elucidate or even quantify how and to what extent this change in decision was determined.

We agree that this is the key primary outcome and we have not described this enough. Reviewer 3 makes a similar point. To deal with this point we have introduced a paragraph in the Methods section. *“The primary outcome measure in this study is the change in MDT decision making. This was determined by each MDT. At each MDT meeting the project lead would review the decisions made for every patient with the rest of the team and decide whether this decision had differed from their normal practice due to COVID.”*

We agree that this outcome measure has a degree of subjectivity and mention this as a limitation to our study in the Discussion section. We have included the sentence *“Our primary outcome measure was, by its nature, pragmatic. This introduces some subjectivity between units as to what they defined as a change in management. As most of these changes involved no surgery or a change in surgical intent, we believe project leads reported significant changes in management that this study sought to explore.”*

The nature of this makes quantitative measures very difficult. Figure 1 shows that there is no clear correlation with this decision making and workload – with Group B had high numbers of patients discussed but low number of changed management.

Minor points:

- p.11, l.27 there is no correlation found between changes in initial management and neither number of patients discussed, nor with number of operations. A bird's eye view of table 1, however, does at least suggest a correlation between the latter two, thus suggesting that the decrease in number of operations is a (logical) result of a decrease of patients presented. Was the 10% change in decision too small to be reflected in a change in operations ? Other explanations ?

I am not fully sure I understand this question. As you would expect there would be a correlation between number of patients discussed and number operated. The variation in changes in decision were due to other actions units did. That is why we used the groupings from the funnel plot.

- it is suggested that, at least for Group B, use of the private sector could achieve allocation of sufficient oncology lists. It would be interesting to see to what extent this solution was used and could the authors explain whether only non-oncological patients were referred to the private sector or also oncological patients ? This should also be discussed in the wake of data from the COVID-19 period indicating that cancellations were less likely to occur in the private sector (Jean e.a., Acta Neurochir 7th April 2020).

This was used for all patients from this unit – including oncological patients. I have now added to this in the results with the sentence – “[using the private hospital sector]... *to provide space to treat brain tumour patients*”. I have used the helpful reference and mentioned this in the discussion in paragraph 3 – the end of that paragraph has the following sentence: “*Our findings complement other studies that have shown that the use of the private sector during this COVID pandemic results in less disruption to services.*[11]”

- Finally, could the authors speculate or quantify on the reasons why no surgery was performed in the 90 cases in whom this normally would have been recommended ? Did additional factors like age or condition also play a factor, was this decision always made because of lack of theatre capacity ?

We have some information on this from the project leads comments. I have now added a couple of sentences to Paragraph 2 of the Results: “*Information from project leads suggest the majority of these patients were elderly and had poor performance status. Other reasons for not operating were patients with low-grade gliomas or meningiomas whose surgery was deferred until after the critical phase of the pandemic.*”

Reviewer: 3

Reviewer Name: Kiran Turaga

Institution and Country: University of Chicago, USA

Please state any competing interests or state ‘None declared’: None declared

The authors provide an analysis of neurosurgical units and describe the change in the MDT management for cancer patients due to the COVID crisis. They further break down units based on the

number of cases or competing interests and find that the differences in management were further amplified.

This is an interesting paper, but the research could have delved further into the question at hand.

1. For instance, if the management was changed in 10% of patients, did that lead to different outcomes for patients, or were they identical? Given the short follow up, I understand not having survival outcomes, but perhaps resectability or completeness of resection rates?

This study was not designed to answer such questions. We do have a little information on whether COVID impacted on this – only 15 patients (11.5% of those whose treatment was changed) had a change in extent of surgery.

2. I think clear understanding of local factors is important to understand the implications of the analysis. Firstly, were MDTs conducted virtually or in person at all institutions? Were all physicians involved or were certain demographics of physicians excluded (eg >65)? Was there an association between testing rates, covid positivity rates, healthcare worker positivity, ICU bed shortage and PPE usage to the recommendation? The authors comment for instance group A had less covid cases, but data would be welcome.

The first part about how the MDTs were conducted is important. We actually collected that data and have now included this in the RESULTS section. I've added two sentences: *"In all units the functioning of the MDT changed with 8 units (33%) moving to fully video conferencing, all of the other units would limit attendance to a smaller number of senior representatives of different specialities (e.g. neurosurgery, oncology, radiology etc). 4 units (17%) reduced the duration of the meeting."*

As for the differences in COVID rates –this was not possible to measure accurately due the the large geographical catchment areas for the neurosurgical units that make assessment of incidence extremely difficult using the published data available. I think the reviewer has misunderstood my description of Group A – it isn't that they had lower COVID positivity rates (we didn't measure this) but the study lead at those units commented that their department was less affected by COVID. I've clarified this by adding new *"brain tumours"* to explain the lower numbers of patients, and have changed the next sentence to start with *"Project leads commented that COVID had less impact on their units or..."*.

The key point about the variation in incidence was that our survey period corresponded to the peak of cases in every region. We have alluded to this in the discussion where we have added, in the limitations about regional variation *"The dates of our survey, however, correspond to the peak in cases for every geographical region"*

3. Are all patients managed according to a pre-specified pathway and there were 10% deviance from the pathway, or was this a subjective assessment of change from what might have been done if not for COVID?

This was a subjective decision that all MDTs made about how they would manage individual patients. Please see reviewer 2, as this is similar. I have outlined this in more detail in the methods section.

4. Can the group include the neurosurgical units from London too? This would give them a control of

practices in a higher incidence zone and may support their suggestion that a lower incidence of covid leads to less change.

As this data was collected prospectively we would not be able to get this data now. The idea of different incidence zones is really interesting and is something we did try to look at but found it not possible for reasons given in the response to Q2 above.

VERSION 2 – REVIEW

REVIEWER	René H.M. Verheijen UMC Cancer Centre Utrecht, Dept. of Gynaecological Oncology
REVIEW RETURNED	04-Jul-2020

GENERAL COMMENTS	The authors have adequately and to the best of their abilities reacted to the issues raised by the 3 reviewers. My verdict that the study design is 'not appropriate to answer the study question' merely reflects the fact that, as discussed by the authors, the study parameters (i.e. decision to change treatment) are subjective and not measurable. As explained by the authors, this is inherent to this type of study.. Otherwise the results of this study are very informative.
--

REVIEWER	Kiran Turaga University of Chicago, USA
REVIEW RETURNED	20-Jul-2020

GENERAL COMMENTS	The authors have attempted to address my previous concerns. I understand the limitations of the dataset. I am concerned about the implications of posing this study as an original research article, which is fraught with several biases, such as the subjective change in management pathways, the heterogeneous changes in tumor board composition, the subjective assessment of how much COVID affected their unit. This would make their study not reproducible, and certainly difficult to infer from. However, in the time of this unique pandemic, this paper has relevant information about the real world experience seen by neurosurgical units in the UK. For that reason, I would support publication of this article as a special report/brief report or a journal format that does not indicate a rigorous scientific basis but rather an article of interest to the readership.
--

VERSION 2 – AUTHOR RESPONSE

RESPONSE TO REVIEWERS COMMENTS - R2

We understand reviewer 3 comments about the experimental design. We would have loved a more detailed study that provided patient-level data, but such a study would have been impossible to collect in a rapid timescale during the middle of a pandemic. We agree with the reviewers comment that our data provides 'real-world data' that shows the impact and, we believe, shows potential ways to abrogate any second wave of infection.

To respond to this we have extensively changed the beginning of the paragraph in the discussion explaining the limitations of our study (Page 13, Discussion) it now reads:

“The study does have limitations principally in the design. The purpose was to provide a prospective, rapid ‘snapshot’ of changes of MDT decision making during the peak of the pandemic using high level data. As such we did not include the detail from individual patient data that would perhaps allow better explanation of individual variation between units. Instead our project leads, all busy clinicians dealing with the impact of the peak of the pandemic, were able to obtain data on patient numbers. Although the study may lack some scientific rigor, it nevertheless provides a narrative of the real-world experiences seen by neurosurgical units during the height of the pandemic in the UK.”